# Dietary Acid Load and Cancer Risk: A Review of the Uruguayan Experience

**DOI:** 10.3390/nu15143098

**Published:** 2023-07-11

**Authors:** Alvaro Luis Ronco, Maximilian Andreas Storz

**Affiliations:** 1Unit of Oncology and Radiotherapy, Pereira Rossell Women’s Hospital, Bvard. Artigas 1590, Montevideo 11600, Uruguay; 2Department of Internal Medicine II, Centre for Complementary Medicine, Freiburg University Hospital, Faculty of Medicine, University of Freiburg, 79106 Freiburg im Breisgau, Germany

**Keywords:** dietary acid load, cancer, Uruguay, neoplasms, potential renal acid load, net endogenous acid production, nutrition, cancer epidemiology

## Abstract

Dietary acid load (DAL) is recognized as a risk factor for several chronic disorders, including obesity, diabetes, and osteoporosis. Recent evidence suggests that an elevated DAL, as measured by the validated potential renal acid load (PRAL) and net endogenous acid production (NEAP) scores, could also increase the risk for several cancers. This narrative review summarizes the potential role of DAL in Uruguayan cancer patients and outlines the potentially involved pathophysiological pathways that mediate the role of DAL in both cancer development and growth. Although Uruguay is a developing country, its average diet is a heavily meat-based Western one, translating into a supraphysiological acid burden from diet. In recent years, we have published epidemiologic evidence based on ten case-control studies involving 3736 cancer cases and 9534 hospital-based controls. Odds ratios and 95% confidence intervals were estimated for each interest variable to analyze the association between the exposure levels of DAL scores and cancer, calculated by unconditional logistic regression. In a majority of the cases, the highest DAL scores tended to double the cancer risk as compared to the lowest category. We also found high risks for methionine intake, an acidifying amino acid found in higher concentrations in animal-based foods, which may increase cancer risks at least by a joint action based on the pH and the proliferation enhancing properties of the amino acid itself.

## 1. Introduction

Cancer is a relevant cause of morbidity and mortality worldwide, and the exposure to risk factors plays an essential role in the biology of most cancer types [1]. Their etiology is usually multifactorial, including genetic, environmental, and lifestyle factors [2]. The latter factors, such as smoking, alcohol consumption, and dietary habits, are considered significant contributing factors in the etiology of cancer, and they are also the main targets for primary prevention [3].

Western-like dietary patterns that usually involve a high meat intake have received increasing attention due to their association with cancer risk [4]. This might be particularly the case in Uruguay, which, although a developing country, has a high human development index [5] and an average diet centered around meat, with the world’s highest per capita beef intake [6]. Over the past quarter century, the relationships between nutrition and cancer risk were explored in this country, including case-control studies on single foods, food groups, nutrients, and dietary patterns [7,8,9,10,11,12,13,14,15,16,17]. Following a recent strategic realignment, a new focus was put on the associations between cancer and dietary acid load, which is a novel marker of diet quality that estimates a diet’s overall acid-based impact on the human body.

### 1.1. Dietary Acid Load

Blood pH is continuously maintained within the range of 7.35–7.45 and tends to be meticulously controlled by the body’s buffer systems to avoid acidosis (a pH of <7.35) or alkalosis (a pH of >7.45). Minimal changes are expected in the value of plasma bicarbonate and blood pH within the normal range [18]. When the pH is balanced at values closer to the lower limit of the normal reference range (e.g., 7.35), this condition is called low-grade or subclinical metabolic acidosis. While various factors can lead to subclinical metabolic acidosis, the consumption of acidifying high-protein Western diets has been identified as a major contributor [19].

Dietary acid load (DAL) is not only determined by protein intake but also by a diet’s mineral content [20,21]. As discussed later in this review, there are two widely used and validated estimates for DAL: the potential renal acid load (PRAL) and net endogenous acid production (NEAP). Both formulas use diet composition data to estimate DAL. Positive overall PRAL scores (>0 mEq/day) indicate a net acidifying diet, whereas negative overall PRAL scores (<0 mEq/day) indicate an alkalizing diet [22]. 

In the past, epidemiological studies have often examined the complex interrelationship between DAL and adverse health outcomes, mainly related to chronic non-communicable diseases, such as osteoporosis and cardiovascular disease [19]. It is now widely accepted that a prolonged diet-induced low-grade metabolic acidosis state may predispose an individual to metabolic abnormalities, such as insulin resistance, diabetes, high serum triglycerides, and obesity [23].

### 1.2. Dietary Acid Load and Cancer

The aforementioned molecular imbalances may indirectly serve as intermediary or downstream effectors of carcinogenesis or tumor promotion [24]. An acidic cellular microenvironment has been shown to impair the removal of DNA damage induced by carcinogens in cultured pulmonary epithelial cells [25]. There is evidence that an acidifying microenvironment promotes the survival of cancer cells and facilitates tumor invasion as well as metastasis, whereas an alkalizing microenvironment has opposite effects [26,27].

In light of these findings, several research groups have recently started to examine potential associations between cancer and DAL, and three independent meta-analyses now suggest that DAL could play an important role in various cancer types [28,29,30]. Amongst others, positive associations with high DAL have been reported with regard to breast [31], colorectal [32], and pancreatic cancers [33], as well as with gliomas [34].

This research has recently also been expanded to include certain amino acids which also contribute to acid production in the human body [35,36,37]. Among these, cysteine and methionine stand out as their catabolism generates not only protons but also sulfates, which are acidic non-metabolizable anions. Foods of animal origin, such as meat, fish, and cheese, are rich in these amino acids and are thus important DAL determinants [38].

Studies on DAL and cancer have been conducted in different countries and on populations, with a noticeable amount of studies in the United States [31,33] and Iran [32,34]. The largest body of evidence, however, comes from a large Uruguayan cohort, and it has been an intensive subject for our research group over the last three years [39,40,41,42,43,44,45,46,47,48]. The major purpose of this review was to summarize this evidence and synthesize these findings for the local Uruguayan context. Here, we describe our study cohort in detail, highlight its most important features, and discuss cancer-specific risks associated with DAL. Finally, we review molecular pathways that are potentially involved in the complex interrelationship between DAL and cancer.

## 2. Case-Control Studies in Uruguay

### 2.1. The Study Cohort

Case-control studies in Uruguay have a long tradition, and the investigated Uruguayan population is characterized by specific features that make it particularly interesting for epidemiological investigations. Our studies are largely based on a large multisite epidemiologic research project which was conducted in Uruguay between 1993 and 2005. 

In the aforementioned period, all newly diagnosed and confirmed cases of cancer registered at the major public hospitals of Montevideo, Uruguay’s capital city, were considered eligible for this study. During this time, four of the city’s leading hospital institutions (Hospital Pasteur, Hospital Maciel, Hospital de Clinicas, and Instituto Nacional de Oncologia) received a relevant proportion of patients from the public health system for cancer diagnosis and/or treatment. Back then, Montevideo centralized its public health system, and approximately 50% of all incident cancer cases were diagnosed there. According to Uruguayan laws, patients admitted to public or university hospitals—usually resource-deprived individuals with low incomes from around the country—had free access to most medical services.

Within the 12-year study period, 3736 cancer cases were identified and included in a large database which we used to build our later published epidemiological studies [39,40,41,42,43,44,45,46,47,48]. Patients hospitalized for conditions unrelated alcohol or tobacco intake and without recent dietary modifications were considered eligible as potential controls. These patients were also pooled from the same institutes during the exact same time frame. A total of 9534 patients who completed the questionnaire were included in the study as controls. The total study sample thus comprised more than 13,000 individuals. Cases and controls were frequency-matched by age (10-year groups), residence (urban/rural), and region (Montevideo/other counties). 

The study was conducted with the local support of specifically trained social workers. The study personnel was not familiar with the study aims and objectives and worked at the hospitals in two phases: First, they performed routine screenings for newly diagnosed cancer patients, working with medical records personnel’s collaboration. Second, they contacted patients who were eligible to be matched by the aforementioned criteria. All participants underwent in-person interviews at their respective hospitals after consenting to the study. Proxy interviews were not permitted as per our study protocol.

### 2.2. The Employed Questionnaires

Participants were requested to answer a structured study-specific questionnaire that included socio-demographic variables, details on occupation, a structured cancer history for first- to second-degree relatives, self-reported anthropometric data for the five years before the interview, smoking and alcohol intake, and a history of “mate” tea and coffee drinking habits. The questionnaire package also included a food frequency questionnaire (FFQ) that encompassed 64 items and was representative of the Uruguayan diet at this time. The FFQ focused on food consumption habits for the five years before the interview. Again, proxy interviews were not accepted. The FFQ was tested for reproducibility [13], allowing the estimation of individual energy intake. All dietary questions were open-ended. Local Uruguayan tables of food composition were used to estimate energy and nutrient intake [49], and an external source was employed to estimate the methionine content in foods [50].

### 2.3. Dietary Acid Load Estimation

We estimated DALs with validated formulas [20,21] and applied them consistently in our ten published epidemiologic studies on DAL and cancer (PRAL and NEAP). The calculations were performed as follows:PRAL (mEq/day) = (0.49 × total protein (g/day)) + (0.037 × phosphorus (mg/day)) − (0.021 × potassium (mg/day)) − (0.026 × magnesium (mg/day)) − (0.013 × calcium (mg/day))
NEAP (mEq/day) = (54.5 × protein g/day))/(0.0256 × potassium (mg/day)) − 10.2

PRAL scores take into account the intestinal absorption rates for various nutrients (protein, potassium, calcium, magnesium, and phosphate) and have been validated against urine pH levels in healthy adults [51]. The rate of sulfuric acid production from protein metabolism and bicarbonate generation from the metabolism of the intestinally absorbed potassium salts of organic acids are prominent, and these become the highly variable components of the NEAP score as developed by Frassetto et al. [21]. These authors stated that “by considering both, the acidifying effect of protein and the alkalinizing effect of potassium (organic anions), NEAP can be predicted confidently from the [...] available contents of only two food nutrients”. The authors also reported a high correlation between the two scores (r = 0.84, *p* < 0.001).

### 2.4. Nutrient Intake Estimation

An analysis program was compiled to calculate energy as the sum of all individual values. Each one was obtained after multiplying the number of servings/year by the ratio of calories of the serving/100 g of each, divided by 365 days. Most typical or average servings of solid foods were within the range of 100–150 g, and this applied to methionine and other nutrients [49,50]. The animal-based nutrients were calculated by adding estimations from all animal foods, and the plant-based nutrients were derived by subtracting the animal-based total from the total intake. For research purposes, a nutrient density was calculated and expressed as the number of daily milligrams or grams of the substance/1000 kilocalories.

### 2.5. Statistical and Epidemiological Analyses Practices

Most questionnaire variables were originally continuous. For a majority of our studies, however, we categorized them for analytic purposes when necessary. In the original studies, the categorization into tertials, quartiles, or quintiles was performed according to the sample sizes obtained. Preliminary univariate analyses were performed to select potentially eligible variables for the later-constructed regression models. For comparison, we selected the interest variables that were presented as mean values ± standard deviations (SDs). We estimated odds ratios (ORs) and 95% confidence intervals (CIs) for each interest variable to analyze the association between the exposure levels of the acid load scores and cancer risk, and these were calculated by unconditional logistic regression.

In all our studies, the reported p-values were two-sided, and associations with *p*-values of <0.05 were considered statistically significant. The regression models were compared using a term for energy as a continuous and categorical variable, searching for the best OR estimates. Potential observable confounders were included in the multivariate analyses. The logistic regression equations included terms for the following independent variables: age (categorical, five), sex (binary), family history of cancer for first- and second-degree relatives (binary, no/yes), smoking status (categorical, three), smoking intensity (pack-years, continuous), and intakes of energy (continuous), total fiber (categorical, five), α-carotene (continuous), lycopene (continuous), calorie-adjusted total iron (continuous), and methionine (categorical, five). All calculations were performed with STATA statistical software (Version 10).

## 3. Results

Table 1 lists the ten case-control studies on DAL and cancer risk conducted in Uruguay in recent years. For this review, we extrapolated the following tumor site-specific data: the numbers of cases and controls, the highest ORs achieved for PRAL and NEAP scores, and the *p*-values for the trends.

Compared to the reference categories, the ORs corresponded to the highest categories (tertials, quartiles, or quintiles). The regression models differed from study to study, and only the study on the whole sample [48] included the model described earlier. Nevertheless, all results corresponded to the most demanding models in terms of model specification techniques. Most tumor sites yielded significant risk increases for high DAL scores. Regarding both scores, there were two sites without a significant risk association: the oral cavity and kidneys.

Table 2 displays the mean PRAL scores and their standard deviations according to the selected sociodemographic variables in the examined Uruguayan study population. The PRAL scores tended to decrease as age increased by decade, particularly among the controls. The cancer cases who were residents of rural areas showed notably higher PRAL scores than those of the participants living in urban areas. The numbers of other counties (where rural areas were prevailing) partially reflected this when they were compared to Montevideo (essentially urban): their respective mean scores for the cancer cases were approximately 1.5-fold higher (7.6 vs. 5.0). Regarding sex, the men’s PRAL scores were twice as high as those of the women for both the cancer cases and the controls. People with lower education years tended to have higher PRAL scores. Considering the body mass index measures, the obese subset unexpectedly did not show the highest scores. The family history of cancer did not reveal remarkable differences: the cancer cases had higher PRAL scores than the controls in the presence and absence of a family history of cancer. Finally, the scores displayed similar behaviors regarding some of the selected habits of the population, such as smoking, alcohol drinking, and “mate” drinking: the more intense these habits were, the higher the PRAL score.

Detailed information about the mean PRAL and NEAP scores for each tumor site and by sex is displayed in Table 3. Among the men, the highest DAL scores were found for the cancers of the upper aerodigestive organs, and the lowest scores corresponded to prostate cancer. Among the women, the highest scores corresponded to breast cancer, whereas the lowest were for kidney cancer. Here, the PRAL scores were significantly lower in comparison to the controls, and the overall association was not significant. In fact, negative values were found in some cases (−2.0), reflecting an alkaline diet. A comparable trend was found for the NEAP scores. Finally, when glancing at the cases and controls in the whole sample, significant differences for the PRAL and NEAP scores between the sexes were observed.

Figure 1 displays a summarized version of the data described in Table 3, with the mean PRAL and NEAP scores presented in descending order. Both images show strong similarities and few differences, and the most notable difference is likely the position of breast cancer in each ranking. As a general trend, the highest mean PRAL scores (above the global average) were found for cancers of the upper aerodigestive tract and the lowest were found for cancers of the abdominopelvic organs. Concerning the NEAP scores, we observed some similarities, but the rankings were not identical.

Due to the close interrelationship between DAL, cancer, and methionine, we recently focused on the latter in an overall whole-sample comparison [48]. Table 4 displays the estimated mean daily methionine intakes as analyzed by sex and by case/control status. Two adjustments were made: (I) by energy (expressed in mg/1000 kcal/day, and (II) by body weight (described in mg/kg weight/day). The cancer cases always showed statistically significant higher methionine intakes than the controls. In addition, some sex-specific differences were found, e.g., the mean methionine intake was significantly higher for women when adjusted by energy, but there were no significant differences when comparing body-weight-adjusted intakes.

A separate sub-analysis of the methionine intake among women was performed (Table 5). The intake was categorized in quartiles, highlighting the following trends: (I) methionine intake showed an opposite distribution where the intake trend decreased among the controls and increased among all the cases; and (II) breast cancer cases showed an increasing intake trend, but there was a marked reduction in the lowest quartiles (~21%) and a marked rise in the highest quartile (35%) compared to the other female cancers (~28% and ~20%, respectively).

## 4. Discussion

The herein presented population sample, gathered over more than one decade to perform several case-control studies, was characterized by an acidogenic dietary style. With the exception of kidney cancer and oral cavity cancers, the cancer cases yielded significantly higher DAL scores as compared to the controls. When the DAL scores were classified in descending order and compared to each other, the cancer sites ranked nearly similarly (Figure 1). The resulting question is: what lessons may be learned from our data? Here, we review trends that were noteworthy across all sub-analyses and which warrant further investigation. We also discuss the potentially involved molecular mechanisms that may play important roles in the association of DAL with cancer.

### 4.1. DAL Scores and Histopathological Features

As displayed in Figure 1, the highest PRAL and NEAP scores corresponded mainly to the upper aerodigestive tract cancer cases, suggesting a higher sensitivity to DAL of the squamous epithelia. The lowest mean DAL scores were primarily found in organ cancers where the glandular epithelia prevailed, such as the lower digestive organs, the breast, and nephrourinary organs. These corresponded to predominantly squamous-type carcinomas and predominant adenocarcinomas, respectively. Considering this scheme, breast cancer was an exception since it displayed one of the highest mean NEAP scores. As regards lung cancers, they formed a substantial fraction of the squamous cell carcinomas, but adenocarcinomas also constituted a relevant proportion globally. Furthermore, the global trend showed an increase in this cell type in parallel with a higher incidence of malignancy among women worldwide. Further research is warranted to clarify whether DAL is associated with a preference for some specific histopathological tumor features.

### 4.2. DAL and Methionine Intake

Acidogenic foods, that is, foods with a positive and high PRAL value, are abundant in sulfur-containing amino acids, including cysteine and methionine. Methionine is an essential amino acid in mammals, and it is unique among the essential amino acids due to its sulfur atom. In addition to its role as a component of proteins, methionine has a key role in several metabolic pathways in epigenetics (S-adenosylmethionine), nuclear functions (polyamines), detoxification (glutathione), and cellular membranes (phospholipids) [52]. Animal-flesh-based food items generally have higher amounts of methionine and cysteine, although their proportions vary substantially. Plant protein sources (including soy and pea) generally contain lower amounts of these amino acids, with different ratios of methionine to cysteine, especially in cereals [52]. The methionine content of selected foods has been discussed in detail in one of our previous publications (Table 6) [51].

The average daily requirement of methionine is 10.4 mg/kg body weight/day [54]. Both an excessive and too-low methionine intake in the diet can cause adverse effects. Sulfur amino acids can also metabolically affect the gut microbiome, increasing H2S to genotoxic levels [55]. Although we found substantially higher methionine intakes in the cancer cases, the human epidemiological evidence for methionine intake and cancer risk is mixed. One study associated an increased methionine intake with a higher risk of prostate cancer, whereas other studies have found reduced risks for colon cancer and no associations with breast cancer [52]. An explanation for these varied results might be related to the heterogeneous nature of the cancer-type-related pathologies. More recent evidence has suggested that methionine restriction inhibits cancer cell growth and may enhance the efficacy of chemotherapeutic agents [56].

The translational potential of the use of methionine restriction as a personalized nutritional approach is currently subject to an intensive discussion. Historically, however, the addition of sulfur amino acids such as methionine and cysteine in the diet have been associated with improvements in the growth rates of several animal species [57], and thus, it is widely employed. Methionine is usually applied for dietary enrichment in animal diets (e.g., livestock, pigs, chicken, and fishes) with the aim of growth improvement and increases in milk production and muscle mass [58,59,60]. The health consequences and implications for humans are still unclear, and the challenge to identify reliable upper intake limits for humans remains [61].

The alkalizing components of plant foods might have decreased in recent decades compared to the data presented in the typical PRAL tables [62]. In the same sense, the methionine components of meats might be higher than commonly assumed based on nutrient tables given the systematic dietary enrichment with methionine for animals. As a corollary, we might have underestimated the methionine intake in our studies. According to some researchers, Western diets contain methionine at levels many times higher than dietary requirements [63]. Our data suggest that the methionine intake in our sample is far above common dietary recommendations, and thus, it could have contributed to the increased DAL and, subsequently, increased cancer risks.

### 4.3. DAL, Phosphorus, and Magnesium Intake

Another important acidifying component and major contributor to DAL is phosphorus. In many countries, the staple habit of eating highly processed foods substantially contributes to an increased phosphorus intake beyond healthy levels. There is evidence that the cellular phosphate burden from phosphate toxicity [64] is a pathophysiological determinant of cancer cell growth—tumor cells express more phosphate cotransporters and store more inorganic phosphate than normal cells, and dysregulated phosphate homeostasis has been associated with the genesis of various human tumors [65].

Although 550 mg/day of phosphorus is considered an adequate intake for adults in the general population, the average intake in European countries is estimated at 1000–1767 mg/day [66]. The dietary intake of phosphorus has increased substantially over time [64] mainly due to the widespread usage of phosphorus-containing additives in foods and new processing techniques [67]. Our study population sample yielded an intake of nearly 800 mg/day (776 mg/day among the controls), which was high based on the quoted reference numbers.

A diet incorporating large amounts of phosphorus-enhanced foods can add an estimated 600–800 mg to global daily intake [68], and it also constitutes a non-negligible amount of sodium chloride (NaCl). In fact, its intake is reported to be an independent predictor of plasma bicarbonate concentration. NaCl may exert approximately 50–100% of the acidosis-producing effect of the DAL and is, therefore, considered a predictor of diet-induced low-grade metabolic acidosis [69]. Regarding the potential links to cancer, it is noteworthy that a high sodium diet converts renal proteoglycans into pro-inflammatory mediators in animal studies [70], which might also pose implications in humans.

In contrast to phosphorus, magnesium is an alkalizing element in the PRAL formula. Several studies have suggested that alterations to intracellular magnesium homeostasis result in cell membrane modifications, increased oxidative stress, alterations to a cell’s proliferation mechanisms, differentiation, and apoptosis [71]. Additionally, magnesium deficiencies often result in inflammation, the activation of inflammatory pathways, and the increased production of pro-inflammatory cytokines by immune cells [72].

### 4.4. Potential Pathomechanisms Mediating the Association between DAL and Cancer

For our cancer-specific studies, we have performed several past literature reviews that have enabled us to summarize the potential pathomechanisms involved in the relationship between a high DAL and cancer development. These mechanisms are displayed in Figure 2.

As reviewed earlier by Williams et al., mild metabolic acidosis—subsequent to a high DAL—aligns with insulin resistance, which, in turn, leads to hyperinsulinemia in the long run [73,74]. The latter may play an important role in tumorigenesis, and it has also been associated with systemic inflammation and visceral adiposity dysfunction, two conditions that synergistically favor the establishment of a pro-tumorigenic environment [75,76,77]. Chronically elevated insulin levels may contribute to the increased production of reactive oxygen species (ROS), which may damage DNA, thereby contributing to mutagenesis and carcinogenesis [78]. The potential oncogenic effects of supraphysiological insulin due to the abnormal stimulation of multiple cellular signaling cascades and the enhancement of growth-factor-dependent cell proliferation could be even further augmented by elevated IGF-1 levels [79,80].

Diets that yield a high DAL are naturally abundant in protein [81]. Although a 2021 study by Lim et al. did not support an interaction effect between DAL and IGF-1 [82], multiple other studies have shown that IGF-1 increases with a high-protein diet [83,84]. The combined effects of hyperinsulinemia and elevated IGF-1 levels could thus be a significant contributor to cell proliferation in DAL-induced cancer development.

The fact that an elevated DAL has been associated with enhanced cortisol production and increased glucocorticoid secretion may even reinforce this [85,86] as high glucocorticoid levels contribute to insulin resistance and thus to hyperinsulinemia [87]. An elevated DAL also reduces the catabolic degradation of potentially bioactive glucocorticoids [87], thus increasing chronically high glucocorticoid levels even further. While potentially protective in the pre-tumor microenvironment, elevated glucocorticoid levels may promote tumor development when sustained over a long time and, particularly, in the context of chronic inflammation [88]. As reviewed by Khadka et al., prolonged glucocorticoid signaling causes a feedforward loop of the increased release of damage-associated molecular patterns (DAMPs), which increase inflammation and further promote tissue damage. In addition, glucocorticoids may suppress anti-tumor surveillance by CD8 T-cells inflammation [88].

Although still controversial, some studies have also suggested that elevated glucocorticoid concentrations inhibit adiponectin secretion and functioning in humans [89,90]. Adiponectin alters glucose metabolism and improves insulin sensitivity while also exhibiting anti-inflammatory properties [91,92]. Elevated glucocorticoid concentrations could thus contribute to the per se reduced circulating adiponectin levels in metabolic acidosis, which inhibits adiponectin gene transcription in adipocytes [93]. Low serum adiponectin levels have been considered to be permissive for cancer development, and a high DAL could contribute to this via the two aforementioned (potentially synergistic) pathways [24].

DAL is likely to contribute to tumor environment acidification and also increases free proton concentrations [94]. Many solid cancers by themselves generate lactic acid as a product of their upregulated fermentative glycolysis, which results in an acidification of the local tumor microenvironment [95,96]. This acidification is a driving force of cancer development as it exerts toxic effects on normal healthy cells while simultaneously promoting tumor invasion [97,98,99]. The acid microenvironment also promotes genomic instability and “T-cell stasis” thereby impairing immune cell surveillance [100,101].

In addition, abnormal cellular acidosis has been shown to trigger human MondoA transcriptional activity, a glucose-dependent regulator of metabolism, which aids cancer cells in metabolic adaptation and apoptosis evasion [94,102,103]. Metabolic acidosis also favors the phosphorylation of cAMP-responsive element binding protein (cREB), a protein known to enhance cell proliferation and apoptosis evasion, which has potential implications in cancer development and metastasis [94,104,105]. It is also noteworthy that altered pH gradients at the plasma membranes of cancer cells have been identified as a potential mechanism in cancer cell drug resistance [106].

### 4.5. Limitations of Our Database and Future Fields of Study

Our reviewed study series builds on a large database that has strengths and limitations that require a detailed discussion in the context of our findings. As for the weaknesses, we have consistently recognized that the studies included a detailed but non-validated FFQ due to external factors. Nevertheless, the FFQ was satisfactorily reproducible in other studies [13]. Although our research dates back to the mid 1990s through to the mid 2000s, the general dietary habits deserve to be considered relatively stable in the Uruguayan population as a recent study has shown the same meat-based solid pattern [107]. As such, our nutritional assessment continues to provide sufficient validity. In light of this, the typical Uruguayan meat-rich dietary pattern might be perfectly suited to estimate DAL since both are closely linked.

Selection bias was minimized by applying state-of-the-art matching techniques. We matched controls and cases by age and residence in each of the original studies in the current database. The overall database dimension and the availability of various tumor sites are additional assets. An additional strength is that all interviews with participants were performed face-to-face (excluding proxy interviews) by the same interviewers, at coincident data collection times, and at the same institutions. This strategy reduced a potential selection bias which was also limited by the nearly full participation of the identified cases and controls, with rates of approximately 97%, which was favored by the study protocol. In addition, the compiled population was comprehensive from a geographical viewpoint and approximately 50% of the study sample derived from the capital city, Montevideo, whereas the remaining 50% came mainly from rural areas. Another strength was the exclusion of persons who reported previous dietary modifications. Nevertheless, the database also has weaknesses that could have influenced our results.

Neither self nor medical prescription and supplementation were registered correctly in the questionnaire. For these and other reasons, mineral estimations became one of the limitations of the present study since they were based on average serving sizes rather than actual food sizes. In addition to that, a higher number of female participants in the study would have been desirable for comparison purposes. However, several malignancies were quite uncommon in women during the study years, and we are thus limited in generalizing our results to both sexes. Another limitation might arise from confounding by cooking methods and mineral contents in water, which were neither asked nor evaluated. 

Comparisons with other countries and populations in which DAL has been assessed as a potential risk factor remain challenging, as well. Due to the very distinct populations and the large regional differences, we refrained from numerical comparisons.

A final limitation is that we were unable to present data on cancer survival rates and quality of life in cancer survivors. This field is largely unexplored as most investigations in the field of DAL and cancer have focused on DAL as a potential risk factor in healthy individuals, and few studies have addressed the role of DAL in cancer survivors [108]. A series of United States-based studies have investigated the impact of a high PRAL score on breast cancer survivors in the Women’s Healthy Eating and Living (WHEL) study [109,110,111,112]. Wu et al. reported a DAL-associated increased total mortality and breast cancer-specific mortality in a cohort of 2950 early-stage breast cancer survivors [109]. Moreover, the same group reported associations between a high DAL and impaired physical function, with role limitations due to physical health, bodily pain, and reduced general health [111]. Apart from the associations with overall reduced physical health, significant associations were found for elevated plasma C-reactive protein and glycosylated hemoglobin HbA1c levels in breast cancer survivors [112]. Both have been previously identified as potential risk factors associated with cancer recurrence and comorbidities in breast cancer survivors [113,114]. Our own research concerning DAL in breast cancer survivors did not include disease-specific endpoints. Still, it again emphasized the acidifying diet characteristics in most cancer survivors [108]. Despite the lack of other large-scale studies in the field, Wu et al. called for considering DAL scores in dietary guidelines for breast cancer survivors [109]. Also, despite the limited body of available evidence, we believe this to be important and hope for additional investigations into other types of cancer soon.

## 5. Conclusions

Our nationwide epidemiologic studies on DAL and cancer risk in Uruguay found that the highest DAL scores tended to double cancer risk for a majority of cancers. We also found high risks related to an increased methionine intake. The latter may increase cancer risks at least by a joint action based on the pH level and the proliferation enhancing influences of the amino acid itself. Many potential pathways are at the nexus of a supraphysiological acid load, low-grade metabolic acidosis, and cancer development, making it challenging to identify a single mechanism that is likely involved in this complex interplay. Rather, we believe that the synergistic effects of all discussed pathways and metabolic alterations could contribute their own part to a more extensive—and potentially even more complex—picture that might explain our epidemiologically observed associations. Consequently, it is conceivable that a reduction in DAL could be a protective measure to reduce the risk of numerous cancers.

## Figures and Tables

**Figure 1 nutrients-15-03098-f001:**
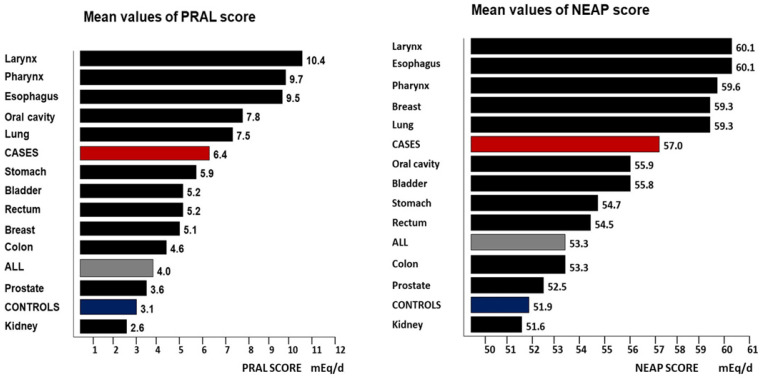
Mean values of the PRAL and NEAP scores stratified by cancer location and shown in descending order. The red color indicates cases whereas the blue color indicates controls. The sample mean is displayed in grey.

**Figure 2 nutrients-15-03098-f002:**
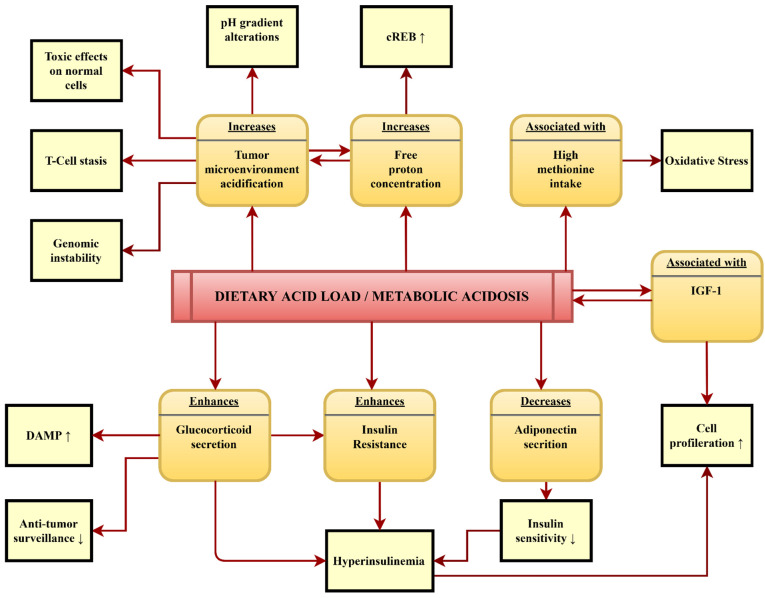
Potential mechanisms associated with an increased DAL that may play important roles in cancer development and growth: a schematic illustration.

**Table 1 nutrients-15-03098-t001:** Summarized data of the Uruguayan studies on DAL and cancer risk, including the highest adjusted ORs found in each.

Ref.	Tumor Site	Cases/ Controls	Cat. ^1^	DAL Score	Highest OR (95% CI)	Trend (*p*)
[45]	Oro-pharynx-larynx	563/563	3	PRAL	2.10(1.46–3.03)	<0.001
			3	NEAP	1.95 (1.37–2.78)	0.002
[45]	Oral cavity	103/563	3	PRAL	1.29 (0.69–2.43)	0.49
			3	NEAP	1.06 (0.56–1.98)	0.98
[45]	Pharynx	185/563	3	PRAL	2.40 (1.44–4.01)	0.005
			3	NEAP	2.28 (1.40–3.71)	0.006
[45]	Larynx	275/563	3	PRAL	2.22 (1.42–3.47)	<0.001
			3	NEAP	2.00 (1.29–3.09)	0.006
[40]	Lung	843/1466	4	PRAL	0.99 (0.64–1.52)	0.94
			4	NEAP	2.22 (1.52–3.22)	<0.001
[46]	Esophagus	185/1110	3	PRAL	2.28 (1.44–3.61)	<0.001
			3	NEAP	2.17 (1.38–3.41)	<0.001
[47]	Stomach	274/1096	3	PRAL	1.74 (1.13–2.66)	<0.001
			3	NEAP	1.90 (1.26–2.84)	<0.001
[39]	Colorectum	611/2394	4	PRAL	1.53 (1.02–2.31)	0.03
			4	NEAP	1.29 (0.89–1.88)	0.11
[39]	Colon	319/2394	4	PRAL	1.29 (0.76–2.19)	0.052
			4	NEAP	1.37 (0.85–2.24)	0.20
[39]	Rectum	292/2394	4	PRAL	1.77 (1.00–3.12)	0.048
			4	NEAP	1.22 (0.73–2.04)	0.24
[41]	Breast	572/889	4	PRAL	2.46 (1.76–3.44)	<0.001
			4	NEAP	1.78 (1.30–2.42)	<0.001
[43]	Kidney	114/864	4	PRAL	0.98 (0.94–1.02)	0.34
			4	NEAP	1.00 (0.98–1.02)	0.92
[44]	Bladder	255/510	3	PRAL	1.74 (1.08–2.82)	0.002
			3	NEAP	1.83 (1.15–2.89)	0.02
[42]	Prostate	323/969	3	PRAL	1.52 (1.01–2.28)	0.01
			3	NEAP	1.72 (1.16–2.48)	0.048
[48]	Whole sample	3736/9534	5	PRAL	2.20 (1.91–2.54)	<0.001
			5	NEAP	2.35 (2.04–2.70)	<0.001

^1^ Number of categories. The *p*-values for the trends correspond to the regressed DAL scores as continuous variables.

**Table 2 nutrients-15-03098-t002:** Mean PRAL scores (±standard deviations) according to the selected sociodemographic and lifestyle behaviors in the analyzed Uruguayan population.

Variables	Categories	PRAL Score: Controls (mEq/d ± SD)	PRAL Score: Cases (mEq/d ± SD)	*p*-Value for Difference
Age groups	<50 years	5.0 ± 10.2	6.9 ± 10.9	0.001
	50–59 years	4.0 ± 9.0	7.4 ± 11.6	<0.0001
	60–69 years	3.3 ± 11.0	6.5 ± 11.2	<0.0001
	70–79 years	2.2 ± 10.5	5.4 ± 11.3	<0.0001
	80–89 years	0.4 ± 12.3	5.2 ± 11.2	<0.0001
Urban/rural status	Urban	2.9 ± 10.6	5.9 ± 11.1	<0.0001
	Rural	3.9 ± 10.3	8.3 ± 11.7	<0.0001
Residence region	Montevideo	3.2 ± 10.5	5.0 ± 10.9	<0.0001
	Other counties	3.0 ± 10.7	7.6 ± 11.5	<0.0001
Sex	Men	3.7 ± 10.7	7.3 ± 11.4	<0.0001
	Women	1.7 ± 9.9	3.7 ± 10.7	<0.0001
Education years	<5	3.4 ± 10.6	6.8 ± 11.1	<0.0001
	≥5	2.7 ± 10.5	5.9 ± 11.6	<0.0001
Body mass index (kg/m^2^)	<18.50	2.1 ± 11.7	8.7 ± 11.1	0.0002
	18.50–24.99	3.6 ± 10.7	6.5 ± 11.1	<0.0001
	25.00–29.99	2.7 ± 10.4	6.1 ± 11.5	<0.0001
	≥30.00	2.9 ± 10.3	6.2 ± 11.7	<0.0001
Family history of cancer for first- and second-degree relatives	No	3.2 ± 10.4	6.6 ± 11.1	<0.0001
	Yes	2.9 ± 11.0	5.9 ± 11.7	<0.0001
Smoking intensity (pack-years)	Non-smoker	2.0 ± 10.3	3.3 ± 11.3	0.0004
	0.01–27.0	2.8 ± 10.8	5.8 ± 11.3	<0.0001
	27.1–53.0	4.5 ± 10.0	7.2 ± 11.0	<0.0001
	≥53.1	3.8 ± 11.0	8.4 ± 10.9	<0.0001
Alcohol drinking	Non-drinker	1.8 ± 10.3	4.3 ± 11.5	<0.0001
	Ex-drinker	0.9 ± 10.8	7.1 ± 12.0	<0.0001
	Current drinker	4.9 ± 10.5	7.9 ± 10.6	<0.0001
“Mate’’ status	Non-drinker	1.5 ± 10.1	3.9 ± 12.2	0.0001
	Ever drinker	3.3 ± 10.6	6.7 ± 11.1	<0.0001
“Mate’’ intensity (liter-years)	Non-drinker	1.5 ± 10.1	3.9 ± 12.2	0.0001
	0.1–39	2.5 ± 11.0	5.8 ± 11.0	<0.0001
	39.1–62.9	3.8 ± 10.3	6.4 ± 10.9	<0.0001
	≥63.0	3.7 ± 10.4	7.7 ± 11.5	<0.0001
Whole sample		3.1 ± 10.6	6.4 ± 11.3	<0.0001

The *p*-values are based on chi-square tests and a Student’s *t*-test.

**Table 3 nutrients-15-03098-t003:** Mean values of the PRAL and NEAP scores for the cancer cases. The units are expressed in mEq/d.

	Men	Women	Men	Women
Tumor Site	PRAL ± SD	PRAL ± SD	NEAP ± SD	NEAP ± SD
Oral cavity	7.8 ± 10.8	-----	55.9 ± 19.0	-----
Pharynx	9.7 ± 11.9	-----	59.6 ± 16.8	-----
Larynx	10.4 ± 11.1	-----	60.1 ± 18.0	-----
Lung	7.5 ± 10.7	-----	59.3 ± 17.7	-----
Esophagus	9.5 ± 10.7	-----	60.1 ± 16.8	-----
Stomach	7.6 ± 11.8	1.9 ± 10.6	57.3 ± 19.2	48.9 ± 14.5
Colon	6.2 ± 11.6	2.9 ± 12.4	54.9 ± 17.2	51.6 ± 16.6
Rectum	7.1 ± 12.5	1.1 ± 12.4	56.3 ± 16.8	50.6 ± 18.3
Breast	-----	5.1 ± 10.1	-----	59.3 ± 18.6
Kidney	4.9 ± 10.2	−2.0 ± 8.5	55.5 ± 15.5	43.3 ± 16.2
Bladder	5.6 ± 10.6	2.7 ± 10.8	56.5 ± 17.5	50.8 ± 11.5
Prostate	3.6 ± 11.7	-----	52.5 ± 16.5	-----
All cases	7.3 ± 11.4	3.7 ± 10.7	57.6 ± 17.6	55.5 ± 18.3
Controls	3.5 ± 10.7	1.7 ± 9.9	52.4 ± 17.9	50.3 ± 16.0
Whole sample	4.6 ± 11.0	2.3 ± 10.2	53.8 ± 18.0	51.8 ± 16.9
*p*-value *	<0.001	<0.001	<0.001	<0.001

All units are expressed in mEq/d. *, *p*-values that correspond to differences between the cases and controls as based on a Students *t*-test.

**Table 4 nutrients-15-03098-t004:** Mean intakes of methionine (±standard deviations) for women and men, as well as cancer cases and controls. The mean intakes were adjusted by energy intake and by body weight.

Subgroup	Units	Controls	Cases	All	*p*-Value for Difference
Men	mg/1000 kcal/day	1724 ± 395	1778 ± 388	1739 ± 394	<0.0001
Women	mg/1000 kcal/day	1784 ± 465	1883 ± 470	1813 ± 469	<0.0001
All	mg/1000 kcal/day	1739 ± 414	1805 ± 413	1758 ± 415	<0.0001
*p*-value for difference		<0.0001	<0.0001	<0.0001	

Men	mg/kg weight/day	38.2 ± 15.4	42.7 ± 16.5	39.5 ± 15.8	<0.0001
Women	mg/kg weight/day	37.6 ± 16.3	43.0 ± 19.3	39.2 ± 17.4	<0.0001
All	mg/kg weight/day	38.1 ± 15.6	42.8 ± 17.3	39.4 ± 16.2	<0.0001
*p*-value for difference		0.11	0.64	0.44	

The *p*-values are based on a Student’s *t*-test.

**Table 5 nutrients-15-03098-t005:** Daily methionine intake, expressed in quartiles, showing the similarities and differences between all the women in the studied sample, the controls subset, the cancer cases, and the breast cancer cases.

Methionine Intake (g/Day)	All Women (%)	Female Controls (%)	All Female Cancer Cases (%)	Breast Cancer Cases (%)	Other Female Cancer Cases (%)
<2.05	1031 (31.1)	795 (33.4)	236 (24.2)	124 (21.7)	112 (27.9)
2.05–2.66	836 (25.2)	606 (25.9)	230 (23.6)	121 (21.1)	109 (27.2)
2.67–3.32	745 (22.5)	520 (22.2)	225 (23.1)	127 (22.2)	98 (24.4)
>3.32	700 (21.1)	418 (17.9)	282 (29.0)	200 (35.0)	82 (20.5)
Total numbers	3312 (100.0)	2339 (100.0)	973 (100.0)	572 (100.0)	401 (100.0)

**Table 6 nutrients-15-03098-t006:** The methionine content in commonly consumed foods of plant and animal origin: an overview.

Food	% Methionine in Protein	mg Methionine/kcal
Almonds	1.15	0.32
Black beans	1.51	1.01
Brown rice	2.26	0.52
Chicken breast without skin	2.77	4.94
Corn	2.11	0.64
Eggs	3.11	2.54
Lean ground beef	2.34	2.05
Lentils	0.85	0.66
Low-fat milk	2.51	1.97
Oatmeal	1.87	0.77
Peanuts	1.23	0.49
Pinto beans	1.51	0.90
Potato	1.61	0.42
Soy (tofu)	1.28	1.36
Tuna	2.96	6.48
Wheat (bulgur)	1.55	0.57

Based on [51,53].

## Data Availability

Not applicable.

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
