# Peer review of "Dietary Acid Load and Cancer Risk: A Review of the Uruguayan Experience"

_nutrients, 2023, doi:10.3390/nu15143098_

Round 1
Reviewer 1 Report
This review article was addressed to evaluate the potential role of dietary acid load (DAL) in Uruguayan cancer patients and to determine associated pathophysiological pathways involved in both cancer development and growth. In this regard, DAL is well known to represent a risk factor for chronic diseases including type-2-diabetes and obesity and, in recent years, Authors have published ten epidemiological (case-control) studies considering 3736 cancer cases and 9534 hospital-based controls. In particular, the association between DAL scores and cancer was estimated, by employing unconditional logistic regression, through evaluation of pertinent variables for which Odds ratios and 95% confidence intervals were calculated. It was shown that the highest DAL scores tended to double cancer risk. Moreover, high risks were found for methionine intake by also its actions on the pH.
Introduction deals with DAL (mostly with reference to Western-like dietary patterns) and relationships linking DAL and cancer. Case-Control Studies in Uruguay were then considered pointing out the modalities of their execution (i.e. the study cohort, the employed questionnaires, the DAL and nutrient estimation as well as the statistical and epidemiological analyses). Results summarize data of the Uruguayan studies on DAL and recurrence of a variety of cancers by also reporting the values of indices for DAL estimation such as PRAL and NEAP (mEq/day). In this respect, selected sociodemographic and lifestyle behaviors were considered together with methionine intakes for women and men. Five tables and one figure were enclosed to illustrate these results. Discussion analyses DAL scores with reference to histopathological features of the studied cancers. Besides, DAL was discussed considering methionine, phosphorus and magnesium intakes. Conclusively, a recapitulatory figure (Figure 2) resumes a series of potential physiopathological mechanisms by which DAL/Metabolic acidosis can affect cancer development and growth. The concluding datum of this review is that a reduction in DAL may represent a significant preventive measure to face the risk of various cancers.
Overall, this review is somewhat interesting in providing significant epidemiological cues to better explain the relationships linking DAL and cancer. The manuscript has been prepared with accuracy as well as lexicon, sentence fluency, “English style”, tables, figures and references are concerned. Only minor editorial refinements, here and there, would be suitable (for instance: line 26: …and the influences of the amino acid itself…=which influences? Please better explain; lines 140/161:…Eestimation…=…estimation).
Only minor editorial refinements, here and there, would be suitable (for instance: line 26: …and the influences of the amino acid itself…=which influences? Please better explain; lines 140/161:…Eestimation…=…estimation).
Author Response
Dear Reviewer #1,
We would like to thank you very much for careful and thorough reading of this manuscript and for the thoughtful comments and constructive suggestions, which help us to improve the quality of this article. All requested changes have been clearly marked in yellow and blue color. We appreciate your input, your advice and the fast peer review. Please find our point-by-point response below. Thank you!
Sincerely,
The authors

Reviewer 2 Report
The article written by Ronco describes “Dietary Acid Load and Cancer Risk: A Review of the Uruguayan Experience.” Overall, this review is well performed. Readers of this journal will be interested in what you have done. After resolving the following comments, I will recommend publishing this manuscript.
1. All tables should state the statistical method used to calculate the p-value.
2. This review focuses on Dietary Acid Load and Cancer Risk in Uruguay. Still, the authors should be desperate to include a few sentences in the introduction about the global trend of this case.
3. Also, a few short sentences are suggested in the Discussion section to discuss how Dietary Acid Load and Cancer Risk differs in Uruguay from the global average
The general standard of English writing in this article is decent, although it requires thorough proofreading to ensure the absence of any grammar, spelling errors.
Author Response
Dear Reviewer #2,
We would like to thank you very much for careful and thorough reading of this manuscript and for the thoughtful comments and constructive suggestions, which help us to improve the quality of this article. All requested changes have been clearly marked in yellow and blue color. We appreciate your input, your advice and the fast peer review. Please find our point-by-point response below. Thank you!
Sincerely,
The authors

Reviewer 3 Report
The authors have examined the potential impact of high DAL levels and dietary intake of sulfur-containing methionine on the occurrence and progression of cancer in the Uruguayan population. They also proposed that squamous epithelia are sensitive to dietary acid load, resulting in squamous-type carcinomas based on DAL scores. Overall, the authors presented the recent findings clearly and concisely. This review article is timely, and I support the publication of this article after minor revisions.
I suggest the authors improve the English quality of the manuscript.
1. Correct the spelling. For example,
Line 121: personnel
Lines 140 and 161: estimation
Line 142: consistently 'in' ten
2. Figure captions could be improved. For example, in figure 1, indicated what the red and blue bars represent in the figure caption.
I suggest the authors improve the English quality of the manuscript.
1. Correct the spelling. For example,
Line 121: personnel
Lines 140 and 161: estimation
Line 142: consistently 'in' ten
Author Response
We would like to thank you very much for careful and thorough reading of this manuscript and for the positive feedback. We regret that there were problems with the English. We revised the language quality during the first round of revisions and believe that it happened that you reviewed the initial (very first) version of this manuscript. As part of the first revision round, the manuscript was edited extensively. All requested changes have been clearly marked in yellow and blue color. We appreciate your input, your advice and the fast peer review. Please find our point-by-point response below. Thank you!
